# CoFee-L: A Model of Animal Displacement in Large Groups Combining Cohesion Maintenance, Feeding Area Search and Transient Leadership

**DOI:** 10.3390/ani12182412

**Published:** 2022-09-14

**Authors:** Nikita Gavrilitchenko, Eva Gazagne, Nicolas Vandewalle, Johann Delcourt, Alain Hambuckers

**Affiliations:** 1Research Unit SPHERES, Behavioral Biology Lab, Department of Biology, Ecology and Evolution, Faculty of Sciences, University of Liège, 4020 Liège, Belgium; 2GRASP, CESAM Research Unit, Department of Physics, Faculty of Sciences, University of Liège, 4000 Liège, Belgium; 3Research Unit FOCUS, Behavioral Biology Lab, Department of Biology, Ecology and Evolution, Faculty of Sciences, University of Liège, 4020 Liège, Belgium

**Keywords:** mechanistic modeling, mean squared displacement, collective movement, individual-based model

## Abstract

**Simple Summary:**

In the current context of climate change and forest cover degradation, the dispersal potential of trees is an issue of great importance. This is particularly the case in the tropics, where trees often need animals to disperse their seeds, as this increases the chances of survival of the progeny and allows migration in the face of climate change. An accurate representation of animal movement is therefore necessary to study the colonization potential of trees in new areas. We have conceived an innovative mathematical model describing the individual movement of gregarious animals, integrating several fundamental features of collective behaviors: cohesion maintenance, feeding area search and leadership. The model was applied to simulate the movements of a wild-ranging troop of primates in a nature reserve. As a result, the model allowed us to simulate the movement of the troop, taking into account the movement and individual characteristics of each member, which is important to consider when the group is composed of many individuals. In the future, this model could be used to improve existing zoochoric seed dispersal models and can be coupled with dynamic vegetation models to predict the shifts of trees species distribution under future climate hypotheses.

**Abstract:**

In the tropics, the conservation of tree species is closely linked to that of animals, as a large proportion of trees are zoochoric and therefore rely on the movement of animals to disperse their seeds in order to increase the chances of the survival of progeny and to allow migration in the face of climate change. Research into the prediction of animal movements is therefore critical but has so far focused only on particular features of collective behavior. In contrast, we included the concepts of cohesion maintenance, feeding area search and transient leadership in a single model, CoFee-L, and tested it to simulate the movement of a wild-ranging troop of primates (*Macaca leonina*). We analyzed and compared observations and simulations with a statistical physics tool (mean squared displacement) and with histograms and χ^2^ (for the step length and turning angle distributions). CoFee-L allowed us to simulate the physical properties of the troop’s center of mass trajectory as well as the step length and angle distributions of the field data. The parametrization of CoFee-L was rather straightforward, as it was sufficient to fix a set of parameters easily observable in the field and then to adjust the values of four parameters that have biological meaning.

## 1. Introduction

It is challenging to understand and describe animal movements in the field of conservation, particularly for the design of conservation areas. The reduction of surfaces devoted to wildlife in order to benefit human activities (e.g., logging, quarrying and sand mining) and the degradation of environmental quality limit the availability of resources and constrain animals in smaller areas. Either these animals are able to adapt their behavior by exploiting the remaining or new resources [1] or their local population eventually becomes extinct [2]. Another challenge in tropical areas is that a large proportion of tree species is zoochoric and needs animals, mainly vertebrates, to disperse their seeds. Seed dispersal, i.e., the movement of seeds away from their parent plant, helps trees escape specific pathogens that accumulate in their vicinity (e.g., [3]) and may provide opportunities to reach new suitable areas in case of changing environmental conditions, notably in connection with climate change. Thus, the conservation of tree species in the tropics is tightly linked to that of animals. Unfortunately, tree species conservation is not considered urgent because they are long-lived and their local extinction proceeds more slowly than in animals. Considering the effect of climate change, repeatedly censused plots in the Andes have highlighted thermophilization, i.e., the upward shifts of lowland, from tree species in warmer areas, a phenomenon observed elsewhere in the tropics [4] and in mountains of temperate countries [5]. Shifts are also perceptible in lowlands at high latitudes [6]. Beyond observational studies, understanding or predicting the consequences of climate change on plant species could be inferred by modeling past, present and future vegetation and plant species distribution and growth. Most often, this is performed with species distribution models and with dynamic vegetation models. However, the projections rely on the implicit assumption of the unlimited dispersal of the diaspores, which is most often unrealistic. The development of a realistic and reliable model of animal movement (to allow for the inclusion of seed dispersal in the future) must therefore be developed [7].

The features of collective animal movements have been intensively studied with individually based models [8,9,10], which consider distinctly each individual interacting with its nearest neighbors to predict individual trajectories. Combined together, those features should allow for a more realistic simulation of collective movements. The first important characteristic that can be highlighted in collective animal movements is the existence of interactions between individuals [11]. Vicsek’s model, which considers animals and micro-organisms as self-propelled particles converting the energy of their environment into directed movement, simulates the basic movement behaviors and clustering of gregarious animals [12]. In this model, individual movements tend to be correlated with those of their neighbors: at each update step, the orientation of each particle in the system is modified as a function of the average orientation of the neighboring particles. Although Vicsek’s model is useful because its minimalism eases the analysis of its predictions, it lacks biological realism. Particles do not avoid collisions nor show attraction to each other, whereas the fact that animals tend to keep a minimum distance from each other and align with their neighbors is a behavior frequently observed in nature [13,14]. New approaches have been developed that focus on the aggregation behavior encountered in biological systems, based on local repulsions, alignment and attraction tendencies between individuals [9,15,16,17,18,19]. These models are generally based on two rules: (1) at all times, individuals try to maintain a minimum distance between themselves, and (2) if individuals do not perform an avoidance maneuver, i.e., do not try to move away from one or more individuals, they tend to be attracted to other individuals to avoid isolation and align themselves according to their neighbors. The second important feature affecting the collective movements is the interaction between individuals and their environment. In order to maintain cohesion during locomotion, gregarious animals need to make collective decisions [20,21]. Many species form complex societies with several levels of communities [22,23,24,25], and the integration of hierarchy allows for an improvement in the realism of animal movement models [26,27]. However, the initiation of a movement may not in some cases not be correlated with the level of hierarchy [28]. The leader is not necessarily permanent; he is in fact very often a leader more due to spatial location than due to hierarchical dominant–subordinate status.

Our objective was then to build and validate a new model (CoFee-L) based on the concepts of statistical physics describing the movements of animals belonging to large groups. We considered together cohesion, i.e., the trade-off between distance among individuals and group dispersal; random behavior in search of food resources; and knowledge of the habitat and social organization, i.e., that the individual closest to food becomes the leader and attracts its congeners until the food is consumed. In contrast to previous studies, the separate concepts they developed are included in CoFee-L in order to come as close as possible to reality. As data for validation, we recorded the movements of a troop of ca. 140 *Macaca leonina* in the wild in a nature reserve of Thailand.

## 2. Materials and Methods

### 2.1. Site

The study took place in the vicinity of the Sakaerat Environmental Research Station, a research station belonging to the Sakaerat Biosphere Reserve, 300 km northeast of Bangkok, Thailand (14°26′ to 14°32′ N; 101°50′ to 101°57′ E). The reserve has an area of 80 sq. km with an altitude ranging between 250 and 762 m asl. It is covered by dry evergreen forest (53.4%), dry Dipterocarpaceae forest (14.8%), old growth forest plantations dominated by *Eucalyptus camaldulensis* and *Acacia mangium* (21.4%), grassland (6.1%), agroforestry (2.6%), bamboo groves (1.5%) and cultures (0.2%). The climate is of a typical monsoonal character with a hot–wet season between May and October, a cold–dry season between November and February, and a hot–dry season between March and April. The mean annual temperature is 25.6 °C, and the mean annual rainfall in the region is 1200 mm (Thai Institute of Scientific and Technological Research, 2017).

### 2.2. Field Observations

The group of interest was a troop of *Macaca leonina* (Blyth 1863) habituated to the human observer and followed in order to study their space use, foraging strategies and seed dispersal effectiveness in a degraded habitat [1,29,30,31]. The troop included between 128 and 153 individuals with 11–15 adult males, 41–48 adult females and 76–90 juveniles in the course of a monitoring period between February 2017 and May 2020. It occupied a home range of 599 ha covered with 78% dry evergreen forest and 22% plantations. The troop was followed from sunrise to sunset for ca. 7 days per month (126 complete days and 35 days interrupted by troop losses, inclement weather, etc.), recording the position of the observer at every minute with a standard field GPS. We assumed that the observer occupied the center of mass (CM) of the troop for the subsequent analysis. The daily trajectories of the troop were 2151 ± 497 m regardless of the observation period.

When the troop stopped at a location and individuals fed for more than 10 min, the zone was considered a feeding area (FA), and an average of the fruiting score for each species present was calculated to characterize the abundance of the FA. The fruiting score P_sm_ of species s for month m is a monthly estimate of fruit production that was made on a sample of referenced trees scattered in the primate home range by visually scoring the fructification intensity (no fruits in the canopy: 0; fruits in 1–25% of the canopy: 1; fruits in 26–50% of the canopy: 2; fruits in 51–75% of the canopy: 3; fruits in 76–100% of the canopy: 4). The more the FA is filled with trees with a high P_sm_, the more productive the FA. In order to characterize the monthly food availability in the study site, and thus to determine periods of food scarcity, a fruit availability index (FAI) was associated with each tree species present in the area [32]. To calculate this index, the following formula was used:$${\mathrm{FAI}}_{m}=\Sigma_{s}\;D_{s}{\bar{B}}_{s}{\bar{P}}_{\mathrm{sm}}$$
where D_s_ is the density of species s (stem/ha), $\bar{B}$_s_ is the mean basal area of species s (m^2^/ha) and $\bar{P}$_sm_ is the average fruiting score of species s for month m.

### 2.3. Trajectory Analysis

Animal moves were characterized with mean squared displacement (MSD). Considering a particle moving in a two-dimensional space, its trajectory can be divided into N consecutive positions recorded with a constant step time Δt during a time period $T=\left(N-1\right)\Delta t$. The MSD measures the deviation of a particle’s position over time in relation to an initial position and is defined (discretely) such that
$${\bar{\rho }}_{\tau }=\frac{1}{N-\tau }\overset{N-\tau }{\underset{i=1}{\sum }}l_{i,i+\tau }^{2}\;,\;\tau =1,\ldots ,\;N-1\;$$
where $l_{i,i+\tau }^{\;}$ is the distance between point i and point i + τ [33]. Since the number of data pairs decreases as τ increases, the uncertainty of the MSD calculation increases. Therefore, τ is usually limited to less than a quarter of the total number of data points (Saxton’s rule, Figure 1) [34].

The MSD is a fundamental tool, as it not only gives an idea of the part of the system explored by a walker but also identifies the type of diffusion one is studying [35,36]. In fact, a particle propagating solely due to diffusion (Brownian motion) will result in a linear dependence of the MSD on time, while a non-linear dependence will be a signature of an anomalous diffusion. The key parameter is the value of α observed in the proportionality relation:$${\bar{\rho }}_{\tau }\;\propto \;\tau^{\alpha }.$$

In the case of α=0, we are simply faced with a stationary process in which no movement is carried out during the observation period. For 0<α<1, the regime is called subdiffusive because the MSD increases less rapidly than in the case of classical diffusion. This kind of situation can be encountered in motion models where there are waiting times between steps or when the spatial domain is restricted. α=1  is the standard exponent between the MSD and time, a characteristic of a diffusive regime. For 1<α<2, the regime becomes superdiffusive and can be encountered in situations in which the length of the steps in a random walk is drawn from a distribution with infinite variance, as in the Lévy Walk. Finally, if α=2, the regime is said to be called ballistic and the MSD increases quadratically with time (Figure 2).

### 2.4. Model Building

In CoFee-L, the movement of a group of particles exploring their environment in search of food is considered as a perpetual succession of three phases. The first phase, called the cohesion phase, allows each particle either to move away from neighboring particles that are too close (Rule 1) or to retrace its steps if it has moved too far from the CM of the group (Rule 2). Rule 1 is the overriding rule, and only one of the two rules can be performed during this first phase. Therefore, if particle A is too far away from the group’s CM but particle B is within its comfort zone, particle A will perform an avoidance maneuver first (and only). If neither of the two rules apply, the evaluated particle does not move. The second phase, called the exploration phase, will allow all the particles to perform a random movement in order to explore their surroundings. The applied motion will be a simple random walk, meaning that the direction of the movements will be completely random. Finally, the third and last phase, called the leadership phase, will allow the group to have directed movement when the particles detect FAs in their environs. More precisely, during this phase, each particle will have the opportunity to check if food is present in its surroundings. When a particle detects a source of food, it then becomes a potential leader (if other particles detect other resources, these particles also become potential leaders). Once all the potential leaders are known, the one that turns out to be closest to its FA acquires the status of global leader of the group, while all the other particles become follower particles, the global leader’s only role being to lead the followers to the detected FA. In the particular case where no FA has been located, no particle acquires the status of leader and therefore cannot lead the group. In order not to leave the particles inactive and to increase their chances of finding a feeding site, a simple random walk is still applied as in the previous phase. Once the third phase is over, the first phase is started again and so on. At the beginning of each simulation, the group of particles is generated around an FA, as the troop sleeps near it [31], and the updates of the particle positions are performed in a random sequence. An illustration of CoFee-L can be found in the Appendix A).

### 2.5. Coding and Parameters

CoFee-L, coded in the C++ programming language, has been developed on a network in order to reproduce as intuitively as possible the application of the three phases described above to a group of particles. The simulation environment is thus composed of cells that can be occupied by a particle, occupied by a food source (FA > 0) or empty. The basic rule is that two particles cannot be on the same cell, i.e., in the same place, and that only particles are allowed to move on the network. CoFee-L is governed by 10 key parameters, the description of which can be found in Table 1 [37,38,39].

To avoid a large number of modifiable parameters, it is necessary to predetermine as many parameters as possible. In particular, *nbrWalker* and *sizeMap* can be set to the number of macaques in the troop (140 individuals) and the size of different FA maps (4500 × 4500 m^2^), respectively. Then, *precisionCM* was fixed to 1 m, allowing the group to cluster sufficiently around an FA before moving on to another one. The parameter delimiting the personal space of each particle (*comfortZone*) was set to 1, which represents an area that is neither too large nor too small in relation to the surface area allocated to an individual (1 m^2^). *radiusTroop* was also fixed and set to *nbrWalker*, as it was observed during the field weeks that the troop could sometimes extend up to 300 m. Finally, we fixed *iterations* because it is wiser to fix this parameter in order to have comparable results between simulations. Furthermore, by setting the number of iterations of the simulations rather than the distance traveled by the group, it randomizes the total length of the daily trajectories, which is more plausible than having fixed daily distances. Thus, from one simulation to another, the group of particles will be generated at different locations on the map and will therefore have a fixed number of iterations in order to cover a greater or lesser distance depending on the FAs encountered on its path. Under different conditions tested, simulated particles need an average of 19,300 iterations to travel approximately 2000 m. With such a value of iterations, the position of the group is thus updated approximately every 2 s since one day of tracking is equivalent to approximately 43,200 s (12 h), which is realistic. The control parameters, *abundanceReach*, *explorationZone*, *velocity* and *levyRatio*, were varied to validate the model and to reveal their impact on the movements and the group’s foraging dynamics.

### 2.6. Model Validation

First, we analyzed the field data via an MSD study to establish the type of movement of the macaque troop under study. Second, we checked how CoFee-L reacted to extreme conditions in order to confirm its basic functioning [40]. To better comprehend CoFee-L, we analyzed and described the simulated trajectories for a set of values of the control parameters. Finally, to set up CoFee-L, we characterized the simulated trajectories with histograms and χ^2^ for step length and turning angle distributions sampled at the same frequency as the collection of field observations. The objective of using the χ^2^ quantity was to provide a method of finding the values of the control parameters that minimizes the differences between the simulated and empirical distributions. The smaller the χ^2^, the better the match between observed and calculated values.

## 3. Results

### 3.1. Field Data

During high food availability in the dry evergreen forest, the troop performs a directed movement (i.e., $\;{\bar{\rho }}_{\tau }\propto \tau^{\alpha }$ with α>1, Figure 3) with the mean *α* = 1.46 ± 0.08. The same types of trajectories are also found for high (resp. low—N.B. resp. for respectively) food availability in plantations (resp. dry evergreen forest and plantations), with a mean value of *α* = 1.69 ± 0.08 (resp. *α* = 1.36 ± 0.15, see Appendix A).

For all other observation periods, the mean exponent *α* varies in relatively the same range independently of the abundance of fruits at the study site (Figure 4).

### 3.2. CoFee-L Flexibility Analysis

A diffusive regime is obtained if the particles are generated in a location free of FAs with the control parameters *abundanceReach* = 0.1 and *explorationZone* = 1 (for this section, the control parameters *velocity* = *levyRatio* = 1 in order to facilitate the understanding and interpretation of the different basic behaviors of the model). These conditions establish an extremely small range for each FA, and the particles are only able to explore the map over short distances. Secondly, a ballistic regime is obtained when the group of particles is generated in an area where FAs are present with the control parameters *abundanceReach* = ∞ and *explorationZone* = 2. In this latter case, in addition to being able to explore the map over larger distances, each particle knows the location as well as the abundance of all FAs (Figure 5).

When *explorationZone* is kept fixed, it can be observed that for *abundanceReach* = ∞, the group efficiently crosses the map in all cases (Figure 6). In contrast, if the memory is small (*abundanceReach* = 5), the group is stuck around its generation location because the particles must randomly explore their environment for a period longer than 19,300 iterations in order to hope to enter the range of an FA. For an intermediate value (*abundanceReach* = 50), the group can either be blocked or take advantage of the FAs in its surroundings, depending on where it is initially generated. One can also notice that the CM does not follow the same trajectory depending on the value of *abundanceReach*. Since the particles can extend up to 140 m around the CM and the second phase of CoFee-L allows for randomization of the motion, different trajectories can be chosen throughout each simulation.

Keeping *abundanceReach* fixed, we observed longer trajectories and more deactivated FAs for higher values of *explorationZone,* except in places too remote with a low density of FAs (Figure 7).

### 3.3. CoFee-L Calibration

Tuning only with control parameters *abundanceReach* and *explorationZone* is not enough to produce realistic trajectories of simulated particles. Although using χ^2^ to compare step length and turning angle distributions between observed and simulated trajectories is generally satisfying for *explorationZone* = 2 and *abundanceReach* values between 5 and 50, trajectories are too short compared to the field data (Figure 8, Appendix A).

To increase the realism, we must tune the control parameter *levyRatio* since large-scale animal movements turn out to be a successive combination of clusters of steps separated by long trips [37]. A satisfying χ^2^ is obtained from approximately *levyRatio* = 35 independently of the abundance of fruits when *abundanceReach* = 0.1 and *explorationZone* = 2 (Figure 9). In this way, the group of simulated particles can perform a random movement (given that *abundanceReach* is very small) up to a certain distance (determined by *levyRatio*), after which *abundanceReach* turns to *∞,* so the group can perform a directed movement. For example, a similar trajectory to the field data for the period of high food abundance in the dry evergreen forest can be observed when *levyRatio* = 35 from an χ^2^, histogram and an MSD perspective (Figure 10 and Figure 11). For high food abundance in plantations (resp. low food abundance in the dry evergreen forest and plantations), *levyRatio* = 60 (resp. also *levyRatio* = 35) gives best results (Appendix A). Variation of the control parameter *velocity* does not improve the quality of simulations regardless of the period of abundance (Appendix A).

## 4. Discussion

In general, animal movement models explain only one aspect of behavior, such as the coherence of the community, the daily variation of movements or the importance of the distribution of food resources [9,12,15,16,17,18,19,26,27]. CoFee-L combines these different aspects regardless of the size of the group studied and simulates the individual movement of each group member, which is important to consider when groups of animals have many individuals that move apart from each other. This allows us to reproduce the physical properties of the troop’s CM trajectory (MSDs), as well as the step length and angle distributions of the field data through a rather simple parameterization, since it is only necessary to fix a set of parameters easily observable in the field (here: *nbrWalker* = 140, *sizeMap* = 4500, *precisionCM* = 1, *comfortZone* = 1, *radiusTroop* = 140 and *iterations* = 19,300) and then to adjust the values of four control parameters that have biological meaning (here: *abundanceReach* = 0.1, *explorationZone* = 2, *velocity* = 1 and *levyRatio* = 35 or 60 depending on the season of observation).

Correlated random walks and levy walks are considered the most optimal strategies in the random search problem [41]. In CoFee-L, if we need to simulate organisms performing trajectories that are similar to those of the levy walk, it is sufficient to set *abundanceReach* to 0.1 and *explorationZone* to 2, find the right *velocity* according to the species studied and look for the right *levyRatio* parameter that minimizes χ^2^ comparing the observed and simulated distributions of angles and segments. In cases where the studied individuals are performing a movement corresponding rather to a correlated random walk, the *levyRatio* parameter can be deactivated and it is then sufficient to search for the correct values of the other three control parameters. In the case of the macaque troop in Sakaerat, they were apparently performing levy walks, as do many other primates [39,42,43,44,45].

The troop in Sakaerat uses four feeding strategies depending on the seasonality of the resources [1,29,30,31]. Macaques rely on a “high-cost, high-yield” strategy during the period of high abundance in plantations by increasing their daily trajectories and home range. During the period of high abundance in the dry evergreen forest, the troop rather displays a “low-cost, high-yield” strategy by intensively foraging in the center of their home range. During the period of low abundance in dry evergreen forest and plantations, the troop finally combines the latter two strategy combinations. High-cost (resp. low-cost) means that individuals make high (resp. low) efforts and thus high (resp. low) energy costs when searching for food. This is characterized for example by an increase (resp. decrease) in the length of daily trips. High-yield (resp. low-yield) means that individuals benefit from a high (resp. low) energy input during foraging due to high (resp. low) nutrient quality and a high (resp. low) abundance of native, exotic and/or human resources. In the simulations, the control parameter *levyRatio* must therefore be used to differentiate between the different periods of food abundance in the movements of the particles in the group. The more it increases, the less the group will be stuck in a place without FAs.

Subsequently, in order to increase the realism of the simulations, several concepts of CoFee-L could be improved. Firstly, the first and third phases of the model could be reconsidered. On one hand, the movement rules ignore dominance hierarchies that are particularly strict in groups such as macaques. The assumption that transient leadership is based on who finds food, and that inter-animal cohesion is linked to its transient leadership, is probably too simplistic. A subordinate animal is unlikely to assume a leadership role for even a short time, and it is unlikely that it will maintain a close proximity to a dominant individual [46]. However, in some cases, the initiation of a movement may not be correlated with the level of hierarchy [28]. In the current form of the model, CoFee-L might be more applicable to ungulate groups adhering to a less pronounced dominance hierarchy. On the other hand, the extent of leadership in relation to followers is not representative of what happens in large groups of wild animals because the followers furthest away from the leader perceive his dominance in the same way as those who are closest. Moreover, each individual in the group has the same weight in the decision-making process. However, there are about 80 juveniles and infants among the 140 individuals considered who almost never take the initiative on the location of a resource [47]. The *cohesion* and *leadership phase* of CoFee-L must therefore be adjusted to increase biological realism. Secondly, CoFee-L does not take into account the fission–fusion phenomenon (as shown in Appendix A). In the case of macaques and other species that form large groups, individuals split into several subgroups for feeding (and thus use several FAs at the same time) and join each other during large movements [48,49]. Departure and/or arrival generation points could then be set up, as some species have particular sleeping sites (e.g., [32,50,51]). In the current context, given the degradation of the environment and low fruit availability, the group was simulated and ended its days around FAs; this was observed in the field to presumably maximize energy intake [31] and also in other species such as the bonobo (*Pan paniscus*) [52]. The edge effect could also be taken into account in the future in order to simulate animal communities evolving in a continuous environment. It is important to develop an approach to limit simulation zones by animal species and thus to spatially limit the action of an animal species on the dispersal of a plant species, using, for example, the data on home range sizes. Finally, it could also be interesting to consider rest periods for the different possibilities of action of the group of particles.

Despite the lack of individual data, CoFee-L is able to realistically reproduce the movement of the CM of the troop. It can therefore be generalized to other types of animals, which is perfectly in line with our future objective of calculating the seed rain they generate. The concept of total seed dispersal kernels (TDKs), i.e., the overall probability of dispersion of a plant individual, population, species or community, combining the influences of all primary, secondary and higher-order dispersal vectors has been reviewed, but few studies have attempted to obtain TDKs for given species. While approaches based on seed rain sampling combined with statistical approaches appear challenging, the identification of fruit-frugivore networks and the most contributing vectors has recently experienced great strides [53]. In this framework, mechanistic modeling, including animal movements, their interactions with the fruiting trees and their physiological requirements, becomes easier to consider, and this type of methodology appears to be able to exploit the available datasets. For instance, the model MOST was built and validated for the seed deposition of the genus *Pourouma* produced by a small group of four golden-headed lion tamarins (*Leontopithecus chrysomelas*) [50]. The animal trajectories in their seasonal home range were analyzed with hidden Markov modeling, which further allows for the generation of state transitions related to local environmental characteristics and accordingly, random steps and turning angles. However, such a model for movement is only appropriate for animal species living in small clusters. It could be upgraded thanks to CoFee-L simulating TDKs for given tree species resulting from the activity of an ad hoc set of animal species, be they living in large groups or not.

## 5. Conclusions

We developed the CoFee-L model to simulate the individual movements of animals living in large groups. This model is quite easy to parametrize with elementary observable information in the field and with four parameters characterizing the behavior of the species. The distribution and food abundance govern the individual movements in the home range. The model could be refined by improving the leadership assignment or adding constraints concerning, for instance, the reuse of sleeping sites.

Using CoFee-L, we sought to improve a model of zoochoric seed dispersal to obtain the combined effect of the main dispersing agents and the TDK of a given tree species. Ultimately, we aim to predict the shift and turnover of zoochoric tropical trees species with a dynamic vegetation model (DVM, e.g., [7,54]) under future climate hypotheses. A DVM is able to compute gross photosynthesis and respiration and to allocate fixed carbon to the short-lived and perennial parts of the plants it simulates, from input data such as monthly climate and atmospheric CO_2_ concentration and traits describing the plant species, such as the specific leaf area and the nitrogen concentration. The DVMs realize transient simulations, i.e., running over time, at several thousands of places. However, DVMs produce only suitability and potential growth for the selected species. By dispersing seeds in the area of interest (for instance with CoFee-L), we could record the annual fate of each seed in terms of germination success or failure, followed by growth, development or death over several decades. However, as the TDK also depends on animal density, which itself is influenced by hunting [55,56,57] and by the loss of areas devoted to forests [58], it appears to be impossible to simulate existing situations. Moreover, simulating a real situation in terms of food availability is impossible due to a lack of data. Therefore, the modeling will allow one to test a variety of conditions of landscape continuity [59] and animal densities in the framework of climate change.

## Figures and Tables

**Figure 1 animals-12-02412-f001:**
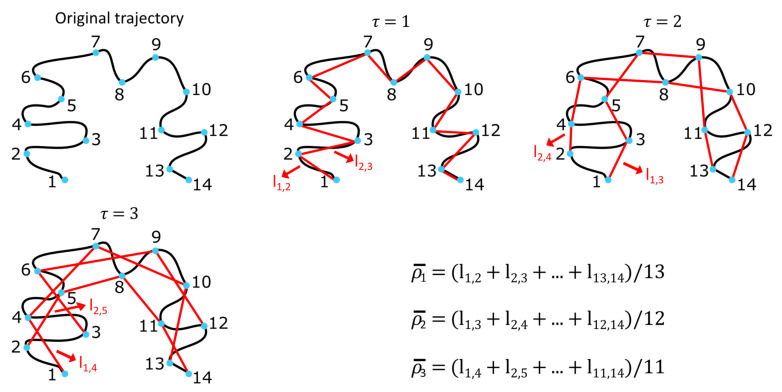
Schematic representation of the mean squared displacement (MSD) calculation for a trajectory comprising 14 points. The MSD is calculated up to τ  = 3, according to Saxton’s rule [34].

**Figure 2 animals-12-02412-f002:**
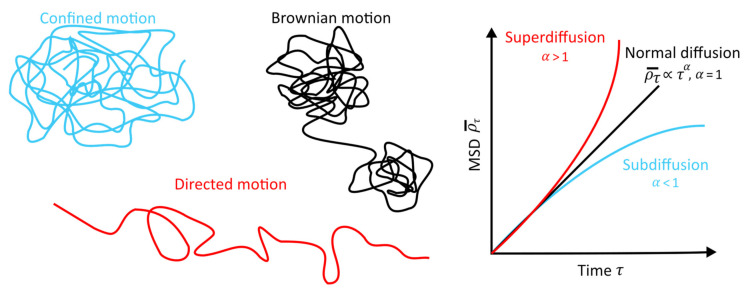
Different types of diffusion will result in different trajectories (**left**), leading to different MSD dependencies as a function of time (**right**). Reproduced with permission from C.H. Menq, Quantitative characterization of cell behaviors through cell cycle progression via automated cell tracking; published by PloS one, 2014.

**Figure 3 animals-12-02412-f003:**
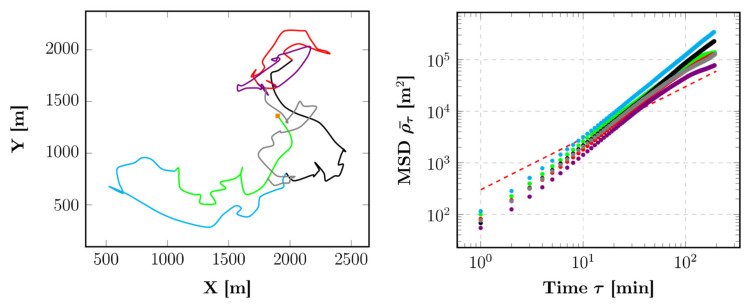
Evolution of the troop position (**left**) with the corresponding MSDs (**right**, mean α = 1.46 ± 0.08) for a tracking of 6 consecutive days during high food availability in the dry evergreen forest. The orange square indicates the departure of the troop, and the dashed line represents a line of a unitary slope, i.e., for α = 1.

**Figure 4 animals-12-02412-f004:**
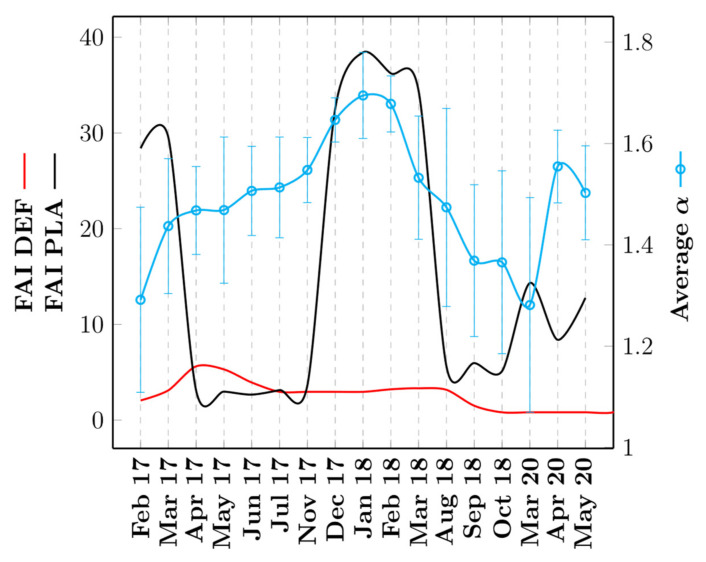
FAI and average α as a function of the different periods of monitoring of the troop. The red (resp. black) curve represents the FAI for the dry evergreen forest (resp. plantations) and is plotted along the left-hand y-axis. The cyan curve represents the variation of the average α and is plotted along the right-hand y-axis.

**Figure 5 animals-12-02412-f005:**
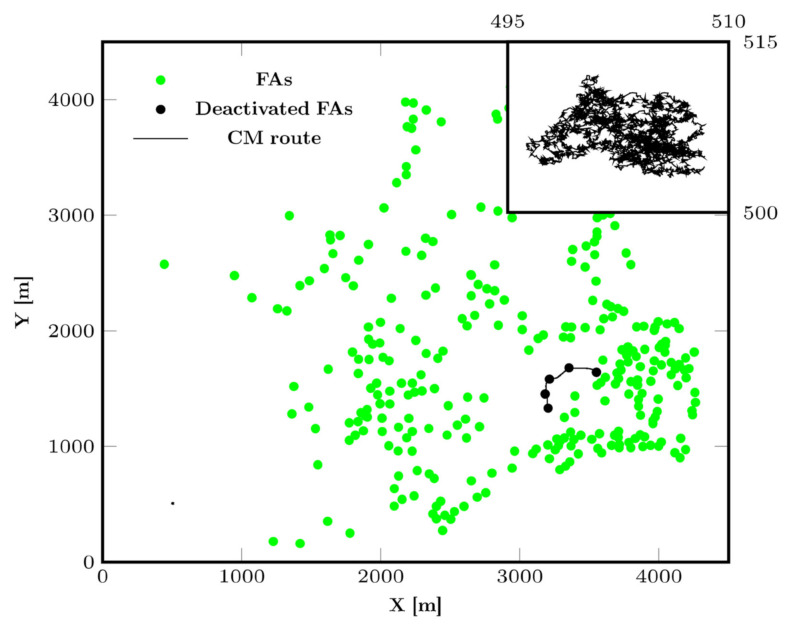
CM trajectories on a map of high food availability in the dry evergreen forest with control parameters *abundanceReach* = 1 and *explorationZone* = 1 (**left trajectory**) as well as *abundanceReach* = ∞ and *explorationZone* = 2 (**right trajectory**). In both cases, *velocity* = *levyRatio* = 1. Deactivated FAs are those through which the group’s CM has passed within *precisionCM* meters. A magnification of the left CM trajectory is shown in the upper right corner.

**Figure 6 animals-12-02412-f006:**
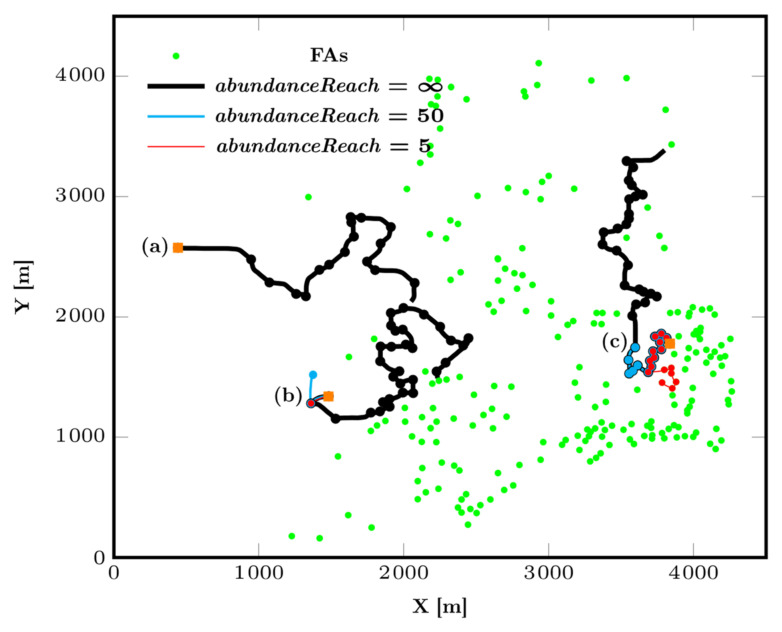
CM trajectories for *explorationZone* = 1 and *abundanceReach* values of 5, 50 and ∞. The orange square indicates where the group of particles was generated. (a,b) The CM remains stuck for *abundanceReach* = 5 and 50. (c) Due to a higher density of feeding area, each value of *abundanceReach* allows the CM of the group to progress.

**Figure 7 animals-12-02412-f007:**
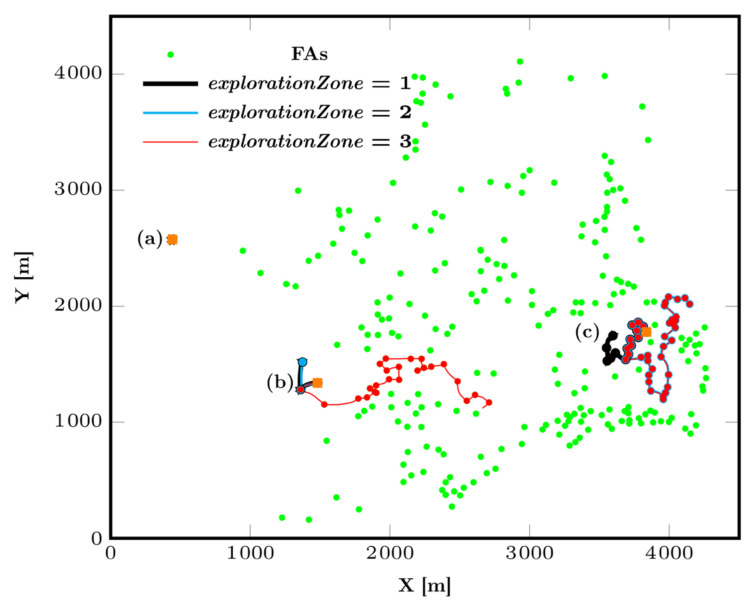
CM trajectories for *abundanceReach* = 50 and *explorationZone* values of 1, 2 and 3. The orange square indicates where the group of particles was generated. (a) The group remains completely stuck at the initial generation point. (b) The particles progress only for *explorationZone* = 3. (c) The density of the feeding area is much higher, and the particles can therefore go in many different directions as soon as the value of the exploration parameter is varied.

**Figure 8 animals-12-02412-f008:**
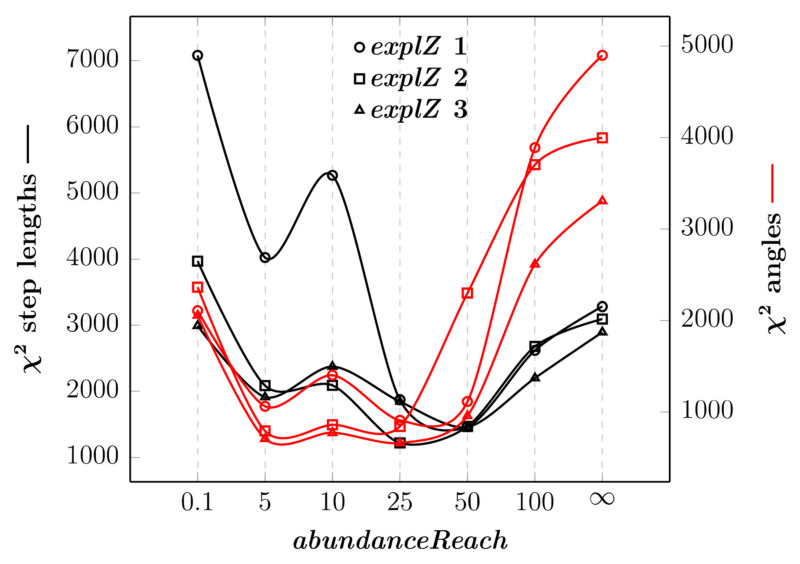
χ^2^ for comparing step length and angle distributions between observed and simulated trajectories as a function of *abundanceReach*, for *velocity* = *levyRatio* = 1 and *explorationZone* = 1, 2 and 3. The black (resp. red) curves represent the χ^2^ for step lengths (resp. angles) and are plotted along the left-hand (resp. right-hand) y-axis. For each simulation, the group of particles was simulated from the same starting point on a map of high food availability in the dry evergreen forest.

**Figure 9 animals-12-02412-f009:**
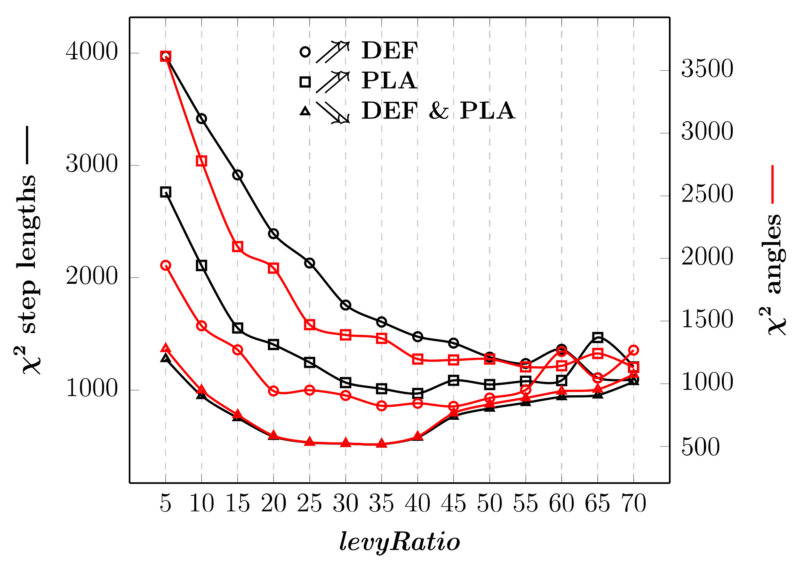
χ^2^ for comparing step length and angle distributions between observed and simulated trajectories as a function of *levyRatio* with control parameters *abundanceReach* = 0.1, *explorationZone* = 2 and *velocity* = 1 (DEF: dry evergreen forest; PLA: plantations). The black (resp. red) curves represent the χ^2^ for step lengths (resp. angles) and are plotted along the left-hand (resp. right-hand) y-axis. For each period of food abundance, the group of particles was simulated from the same starting point at three random positions.

**Figure 10 animals-12-02412-f010:**
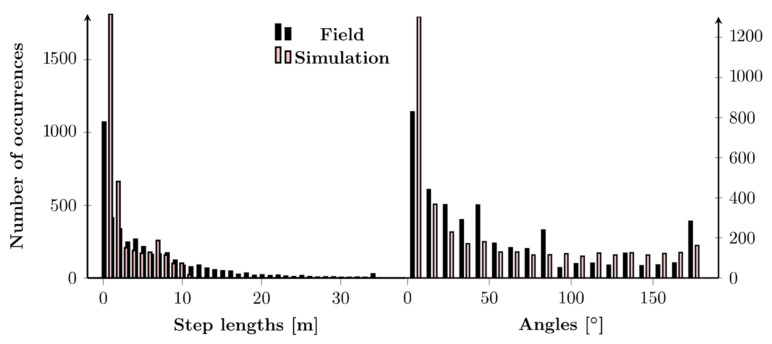
Histogram of step lengths (**left**) and angles (**right**) for field (black) and simulation (red) data during high food availability in the dry evergreen forest, with control parameters *abundanceReach* = 0.1, *explorationZone* = 2, *velocity* = 1 and *levyRatio* = 35.

**Figure 11 animals-12-02412-f011:**
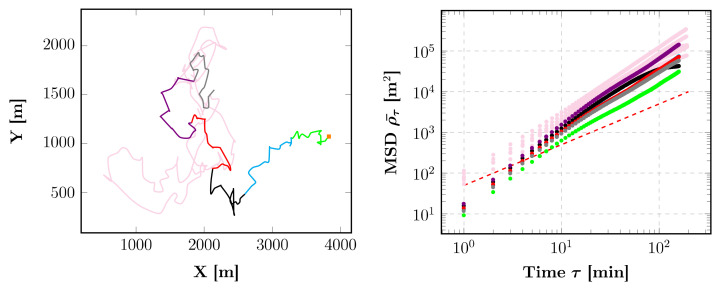
Evolution of the group position (**left**) with the corresponding MSDs (**right**, mean α = 1.53 ± 0.08) for a simulation of 6 consecutive days during high food availability in the dry evergreen forest, with control parameters *abundanceReach* = 0.1, *explorationZone* = 2, *velocity* = 1 and *levyRatio* = 35. Pink trajectories and MSDs represent the troop data. The orange square indicates the departure of the group of particles and the dashed line represents a line of a unitary slope, i.e., for α = 1.

**Table 1 animals-12-02412-t001:** Explanation of the 10 parameters governing CoFee-L.

Parameter	Definition	Details	Example
*nbrWalker*	Number of particles simulated.	/	/
*sizeMap*	Size in terms of cells of an edge of the network.	Since the model simulates particles on a grid, it is appropriate to express some parameters in terms of the grid cell. A cell has an area of 1 m^2^ and from a practical point of view, the simulations were carried out only on square networks.	/
*iterations*	Number of times the 3 phases are performed during a simulation.	The larger the iterations, the more the group’s position is updated and the greater the distance covered by the center of mass (CM).	/
*precisionCM*	Distance in meters at which the group’s CM is considered to have reached a cell with a food source.	A destructive encounter dynamic is applied as in a previous study, i.e., the fruit availability index (FAI) is set to 0 in order to allow the particles to detect and move towards another feeding area (FA). The notion of FA deactivation combines two biological realities, namely, satiety and resource depletion.	*precisionCM* = 10 means that, as soon as the group’s CM is less than 10 meters from an FA, this source is considered reached.
*comfortZone*	Comfort zone in terms of cells of each particle.	Intervenes in the priority rule of the first phase of CoFee-L.	*comfortZone* = 2 means that each particle requires a zone free of any other particle of two cells around it (including diagonals).
*radiusTroop*	Radius in meters around the CM of the group beyond which a particle must return to the CM.	Intervenes in the secondary rule of the first phase of CoFee-L. This parameter allows the group not to scatter beyond a certain distance.	*radiusTroop* = 100 means that particles can move up to 100 m away from the group’s CM.
*explorationZone*	Exploration distance in terms of cells.	Intervenes in the second phase of CoFee-L. The larger this parameter is, the further the particles are able to explore.	*explorationZone* = 2 means that each particle can move two squares around it (including the diagonals).
*abundanceReach*	Range in meters of each FA. The range of an FA is the distance over which it can be detected by a particle. It is defined in CoFee-L as the product of its abundance and *abundanceReach*. The abundance is computed for each FA as the mean P_sm_ for all fruiting trees occupying the FA. Two cases can be distinguished: *abundanceReach* = ∞ and *abundanceReach* > 0. In the first case, all FAs have infinite range regardless of their abundance, and each particle therefore knows the position of all resources. In the second case, the range of each source depends on the value of *abundanceReach*.	Intervenes in the third phase of CoFee-L. This parameter can be seen as the memory of the particles, because the larger it is, the more the particles know the location of a large number of resources, and vice versa. *abundanceReach* combines two biological realities, namely, the individual’s memory of the environment and the remote perception of resources. These two realities form the personal information of an individual.	*abundanceReach* = 100 means that all sources with an abundance of 4 are detectable within a radius of 4 × 100 = 400 m around them, all sources with an abundance of 1.5 are detectable within a radius of 1.5 × 100 = 150 m, etc. (Appendix A).
*velocity*	Velocity of each particle when an FA is attainable.	Intervenes in the third phase of CoFee-L.	*velocity* = 2 means that the particles move by a step of 2 m at each execution of the third phase.
*levyRatio*	Prevents the group from remaining confined by allowing all FAs to be detectable. If any FAs have not been deactivated for 2000/*levyRatio* meters, *abundanceReach* will change to ∞ to “unlock” the group’s progress.	For certain values of parameters such as *explorationZone* = 1 and *abundanceReach* = 1, it is possible that the group remains stuck at a place and cannot evolve any more on the map (since particles cannot explore over a long distance during phase 2 and FAs are not very detectable). The numerator was chosen at 2000 m because it is in the range of the daily average distance travelled by the observed troop.	*levyRatio* = 5 means that the parameter *abundanceReach* will change to ∞ after 2000/5 = 400 m if any FAs have been deactivated during this move. If *levyRatio* = 25, the group will therefore know the position of every FA on the map if no FA has been deactivated after 2000/25 = 80 m.

## Data Availability

The datasets used and analyzed during the current study are available from the corresponding author upon reasonable request.

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
