# Peer review of "CoFee-L: A Model of Animal Displacement in Large Groups Combining Cohesion Maintenance, Feeding Area Search and Transient Leadership"

_animals, 2022, doi:10.3390/ani12182412_

Round 1

Reviewer 1 Report

Dear authors,

Very very good job. Congrats!

Author Response

Dear Reviewer 1,

thank you for your congratulations.

I wish you all the best,

G. Nikita 

Reviewer 2 Report

This manuscript introduces a model that can be used to predict the movement of groups of animals based on food availability, cohesion maintenance, and transient leadership, and tested this with observations on a wild macaque group. The authors in their abstract and introduction emphasize the value of this approach in relation to seed dispersal (zoochory). The paper however, does not demonstrate this with data, but rather suggests that a next step in the model is to use this approach to develop and improve models of seed dispersal in light of climate change-induced habitat alterations. This is a valid goal, but since it is not formally evaluated in this paper, this should be downplayed, and, in this reviewer’s opinion, relegated strictly to the discussion and conclusions as “next steps.”

The paper needs extensive English-language editing (I have included as many edits below as I could but thorough copy-edit is necessary).

The model is complicated, but makes some intuitive sense. I note though, that it might be more applicable to a group of ungulates that does not adhere to as strict a dominance hierarchy as does a macaque group. The assumption that transient leadership is based on who finds food, and that inter-animal cohesion is related to his transient leadership, is likely a bit too much of an over-simplification. A subordinate animal is not likely to assume a leadership role, even for a brief time, nor is a subordinate likely to maintain close proximity to a dominant individual. For example, in section 2.4, in describing the cohesion phase, the movement rules ignore linear dominance hierarchies, which tend to be quite strong in macaques.

I offer some recommendations for improving the readability of this manuscript.

Section 2.5 describes the 10 parameters used to develop the model. This would be much easier if it was placed in a table, with column headings such as “Parameter”, “Definition”, “Details”, and “Example”. It would make it much easier for the reader to keep track of each of these, and/or to refer back for clarification.

I recommend reducing the number of figures if possible – currently there are 14 figures in the paper and 8 more supplemental figures – surely this can be reduced without reducing understanding?

Presumably this is just a distinction in notation, but in the figures (10, 11, 14) that illustrate X2 (step lengths and angles), these are decimal values, correct? It is also quite unclear how the X2 is being used in this context. Presumably it is comparing observed to expected values, but for these figures to have meaning, one needs to specify what the critical value is, or at least indicate where significant differences are and what the DF is. It is possible that I am not understanding how the X2 test was used, in which case further explanation and clarification is warranted.

Some specific wording changes:

P 1 lines 15-16: change need often to often need

P 1 line 21: delete “the” in both cases

P 1 line 22: delete “the” between “and” and “leadership”.

P 1 line 23: insert “us” after “allowed”

P 1 line 29: Change “is zoochore” to “are zoochoric”

P 1 line 31: change “movements” to “movement”. “Cardinal” is not the correct word here; critical? Important?

P 1 line 32: what does “its” refer to (“despite its productivity)

P 1 line 37-38: I would suggest changing “allowed to reproduce the physical properties” to “allowed us to simulate the physical properties” or something like that.

P 2 line 47: delete “the” between “of” and “conservation”

P 2 line 48: change “of the surfaces,” to “of surface”. Clarify what you mean by “in benefit to human activities

P 2 line 52: remove the comma after challenge and after areas. Throughout, I’m not sure if “zoochore” is the correct word; should it be “zoochory?”

P 2 line 57: delete “the” between “of” and “tree”

P 2 line 59: suggest rewording as follows: “…they are long-lived and their local extinction proceeds more slowly…”

P 2 line 66: delete “The” at the start of the sentence.

P2 line 78: suggest rewording as follows: “…should allow a more realistic simulation of collective movements.”

P 2 line 80: Delete “The” before “Vicsek’s”

P 2 line 83: change “a gregarious animal” to “gregarious animals”

P 2 line 87: delete “of” between “lacks” and “biological”

P 2 line 88: change “other’s” to “other” (also on line 89)

P 2 line 97: delete “however”

P 3 line 104: “dominant-dominated” is awkward. Do you mean “dominant-subordinate”?

P 3 line 111: change “was” to “is”

P 3 line 112: delete “the” before CoFee-L

P 3 Line 114: delete “primates belonging to the species”

P 3 line 133: delete “principally”

P 3 line 138: delete “during several field workd”

P 3 line 139: delete “ensuing”

P 3 line 141: change “It is supposed” to “We assumed”

P 4 line 151: should “abundant” be “productive”?

P 4 line 161: Delete “The” in 2 places on this line.

P 5 line 185: delete “a” between “of” and “classical”

P 5 line 191: I believe “said” should be changed to “said to be called”

P 5 line 200: insert “the” between “called” and “cohesion” (also do so before “exploration phase” on line 206 and “leadership phase” on line 209).

P 7 line 289: change “predetermined” to “predetermine”

P 9 line 351: add “an” between “establish” and “extremely”

Figure 12: please label the Y axis more clearly. What are the units?

P 14 line 458: delete “the” between “of” and “behaviour”

P 14, line 494: It is unclear what “Later” refers to?

P 15 line 496: insert “the” between “on” and “one”

P 15 lines 501-502: you note here that the leadership phase does not represent biological reality with respect to the macaque group you used to validate the model. This is quite true, and warrants more extensive discussion. As I mentioned, it might be valuable to test this model on ungulates.

P 15 line 510: include scientific name for bonobo.

P 15 line 526: change “frugivores” to “frugivore”.

P 16 line 550: change “we could” to “we hope to” as this is the next step in further developing your model to predict movements and zoochory.

P 16 line 561: “surfaces” is not the right word here. Perhaps it should be “areas”?

Overall, I think the manuscript presents an interesting model that could have broader application. I encourage the authors to focus more clearly on what they did, and not what they will do. They present no validation of the model with respect to seed dispersal, yet this is emphasized in the abstract and introduction.

Author Response

Dear Reviewer 2,

I wish you all the best,

G. Nikita 

Reviewer 3 Report

Introduction is well written and described gap in knowledge and rationale for study. Computational approach and all related parameters are well described. Figure 7 font is too small and difficult to read. Initializes in Figure 8-10 are difficult to read (are not legible). Recommend bolding fonts for axis labels and legends embedded in figures. All conclusions are supported by results. Discussion for improvement and further study are sufficient and informative.

Author Response

Dear Reviewer 3,

I have modified the figures according to your comments. Thank you.

I wish you all the best,

G. Nikita